# Effects of Cu Addition on Mechanical Behaviour, Microstructural Evolution and Anti-Corrosion Performance of TiAl-Based Intermetallic Alloy under Different Strain Rates

**DOI:** 10.3390/ma14175056

**Published:** 2021-09-03

**Authors:** Cheng-Hsien Kuo, Tao-Hsing Chen, Ting-Yang Zeng

**Affiliations:** 1Department of Mold and Die Engineering, National Kaohsiung University of Science and Technology, Kaohsiung 80778, Taiwan; chuckkuo@nkust.edu.tw; 2Department of Mechanical Engineering, National Kaohsiung University of Science and Technology, Kaohsiung 80778, Taiwan; stars0619@yahoo.com.tw

**Keywords:** TiAl-based intermetallic alloy, strain rate, microstructural evolution, fracture surface

## Abstract

TiAl-based intermetallic alloys are prepared with Cu concentrations of 3–5 at.% (atomic ratio). The mechanical properties and microstructural characteristics of the alloys are investigated under static and dynamic loading conditions using a material testing system (MTS) and split-Hopkinson Pressure Bar (SHPB), respectively. The electrochemical properties of the various alloys are then tested in Ringer’s solution. It is shown that the level of Cu addition significantly affects both the flow stress and the ductility of the samples. For Cu contents of 3 and 4 at.%, respectively, the flow stress and strain rate sensitivity increase at higher strain rates. Furthermore, for a constant strain rate, a Cu content of 4 at.% leads to an increased fracture strain. However, for the sample with the highest Cu addition of 5 at.%, the flow stress and fracture strain both decrease. The X-ray diffraction (XRD) patterns and optical microscopy (OM) images reveal that the lower ductility is due to the formation of a greater quantity of γ phase in the binary TiAl alloy system. Among all the specimens, that with a Cu addition of 4 at.% has the best anti-corrosion performance. Overall, the results indicate that the favourable properties of the TiAlCu_4_ sample stem mainly from the low γ phase content of the microstructure and the high α_2_ phase content.

## 1. Introduction

Intermetallic alloys have attracted significant attention in the literature in recent decades due to their high operational temperatures, good wear resistance, superior hardness, and good electrochemical resistance. Among the many intermetallic alloys which have been developed, titanium aluminium (TiAl) is one of the most widely used in the aerospace and automotive industries due to its light weight and high strength [1,2]. TiAl alloy may exist with two basic microstructures: duplex and lamellar [3]. For both microstructures, the room-temperature ductility and fracture toughness are inversely related. Generally speaking, the creep resistance and fracture toughness of lamellar TiAl alloy are better than those of duplex TiAl alloy under ambient temperatures. However, as the temperature is increased to 800 °C and more [3,4,5], a sudden and substantial loss in the creep resistance occurs.

The literature contains many attempts to improve the room-temperature ductility and high-temperature strength of lamellar TiAl through the addition of substitutional elements (e.g., Nb and Mo [6,7,8]), or interstitial elements (e.g., B, C and N [9,10]). Nb and Mo both serve as β-stabilizing elements, and result in the formation of disordered β-phase with a better ductility than α_2_ or γ phase at elevated temperatures. Meanwhile, interstitial elements, such as C and N, improve the fracture strain and yield stress. Cu-based TiAl alloys are a promising solution for high-temperature structural applications with high strength and toughness requirements [11,12]. Moreover, several studies have shown that Cu addition is also beneficial in improving the biocompatibility, corrosion resistance, and ductility of TiAl alloy [13,14,15].

For many engineering metals and alloys, the flow behaviour under elevated strain rates and temperature conditions is extremely complex. For example, significant work hardening frequently occurs as the strain rate increases [16]. By contrast, a higher temperature usually results in thermal softening [17]. Thus, the flow behaviour of the material is essentially the result of a competition between the effects of work hardening and thermal softening, respectively. It was shown in [18] that the dependence of the flow stress on the strain rate and temperature is associated with the motion of glide dislocations over obstacles in the microstructure. Moreover, under extreme strain rate conditions, the flow response is determined primarily by the effects of viscous drag and/or dislocation generation [19]. The literature contains various constitutive equations for describing the stress−strain response of metals and alloys under different strain rates and temperatures, including the Zerilli-Armstrong and Johnson-Cook models [20,21,22].

As an alloy deforms, the microstructural evolution is dependent on both the strain rate and the temperature. At higher strain rates, dislocation multiplication occurs more readily and prompts a strengthening effect [23,24]. Conversely, at higher temperatures, dislocation annihilation occurs, which leads to a significant loss in flow resistance [25,26]. Both effects have a major impact on the mechanical properties of the deformed microstructure, and must therefore be carefully considered when attempting to predict the response of metals and alloys in typical service applications.

The quasistatic mechanical response of TiAl intermetallic alloys has attracted considerable attention in the literature [27,28,29]. However, the effects of high strain rate deformation on the flow behaviour of TiAl alloy are still unclear. Nonetheless, developing such an understanding is essential since high strain rate deformation often results in the generation of adiabatic shear bands, which then prompt crack initiation [30,31,32,33]. Accordingly, this study investigates the mechanical properties and microstructural evolution of Cu-based TiAl intermetallic alloy at strain rates in the range of 10^−3^~4 × 10^3^ s^−1^ using a material testing system (MTS) and split-Hopkinson Pressure Bar (SHPB). The experimental tests are performed at room temperature using TiAl specimens with Cu contents of 3~5 at.%, respectively. The electrochemical properties of the various samples are additionally tested under ambient conditions in Ringer’s solution over a voltage range of −600 mV to 200 mV.

## 2. Material Preparation and Experimental Procedures

High-purity (99.99%) Ti, Al and Cu powders were purchased from Golden Optoelectronic Co. Ltd. (Golden, Optoelectronic Co. Ltd, New Taipei city, Taiwan). Cu-based TiAl intermetallic alloys were prepared in a vacuum arc melted furnace back-filled with argon. The specimens contained an Al content of 46 at.%, Cu contents of 3, 4 or 5 at.%, and a balance of Ti. To ensure compositional homogeneity, the specimens were heated at 1473 K for 24 h, allowed to cool to room temperature in the furnace, and then reheated to 1473 K once again. Furthermore, for each specimen, the heating and reheating cycle was performed at least three times. The as-cast circular ingots (with dimensions of 30 mm × 4 mm) were machined into cylindrical specimens (length: 5 ± 0.1 mm; diameter: 5.1 mm) using an EDM machine and centre-grinding process.

The mechanical properties of the various samples were investigated at room temperature under both quasistatic and dynamic loading conditions. The quasistatic tests were performed at strain rates of 10^−3^, 10^−2^ and 10^−1^ s^−1^, respectively, using an MTS 810 testing system (MTS Systems Corporation, Minneapolis, MN, USA). The dynamic tests were performed at strain rates of 3 × 10^3^ s^−1^, 4 × 10^3^ s^−1^ and 5 × 10^3^ s^−1^ using a SHPB system (Advance Instrument Inc. Corporation, Taipei, Taiwan). In performing the dynamic tests, the specimen were lubricated with molybdenum disulphide grease to ensure frictionless conditions, and were placed between the incident bar and transmitter bar of the SHPB system. The incident bar was then launched by a gas gun system. The resulting stress–strain response of the specimen was evaluated by measuring the stress waves propagating through the SHPB system using strain gauges mounted on the incident and transmitter bars, respectively.

The microstructural properties of the test specimens were observed by scanning electron microscopy (SEM; JSM-7001; JEOL Ltd., Akishima, Japan) and optical microscopy (OM; Metallurgical Microscopy MR5000, Kaohsiung, Taiwan). In addition, the phases of the specimens with different Cu additions were determined via X-ray diffraction and the scattering angle was from 20° to 120° at speed of 1° per minute and the operation voltage of 40 kV with Cu Kα radiation (Philips X-ray diffractometer; Spectris plc., Almelo, The Netherlands). Finally, the electrochemical corrosion properties of the samples were examined by electrochemical tests performed in Ringer’s solution (9.0 gL^−1^ NaCl, 0.43 gL^−1^ KCl, 0.24 gL^−1^ CaCl_2_ and 0.20 gL^−1^ NaHCO_3_) using a potentiostat (Model 362, EG&G, Instruments, Princeton Applied Research, Princeton, NJ, USA) with a silver chloride electrode (SCE) as the reference electrode. For each specimen, five polarisation curves were obtained under room temperature conditions (28 °C ± 1 °C) using a voltage range of −600 mV to 200 mV and a scan rate of 1 mV/s. To ensure experimental reliability, data collection was under similar conditions and standard deviation is within reasonable range (±10 mv) the polarisation curve was obtained, and each alloy was tested five times, with a new sample being used on each occasion.

## 3. Results and Discussion

### 3.1. XRD Structural Analysis

Figure 1 shows the XRD patterns of the three as-cast specimens. It can be seen that all three samples had a lamellar structure with a mixture of γ and α_2_ phase. As the Cu content increased from 3 to 4 at.%, the intensity of the main γ peak decreased, while that of the main α_2_ peak increased. However, as the Cu content further increased to 5 at.%, the magnitude of the γ peak increased once again. It has been reported that the presence of the α_2_ phase enhances the ductility and strength of TiAl intermetallic alloy, whereas γ phase leads to increased brittleness [34].

### 3.2. Stress−Strain Analysis

Figure 2a–c show the stress–strain response of the various TiAlCu_x_ specimens under quasistatic deformation conditions. For all of the specimens, the flow stress increased at higher strain rates. Moreover, at each strain rate, the maximum stress increased with an increasing strain rate. Overall, however, the results showed that the maximum flow stress was determined mainly by the strain rate. All of the specimens ultimately fractured at high strain values. However, the fracture strain reduced as the strain rate increased (see Table 1), which suggests that the TiAlCux specimens undergo a strengthening effect at higher strain rates. Comparing the various stress−strain curves in Figure 2a–c, it was found that the TiAlCu_x_ sample with 4 at.% Cu addition had the highest strength and fracture strain among all the samples. In other words, the results confirm the XRD finding above that the TiAl_46_Cu_4_ sample has the best mechanical properties of the considered alloys. The effect of strain rate on the maximum flow stress was more than fracture strain, we can see from Figure 2a or Figure 3a, the maximum flow stress of TiAlCu_3_ was increasing from 1500 MPa at strain rate of 10^−3^ s^−1^ to 2100 MPa at strain rate of 4 × 10^3^ s^−1^. The flow stress was increasing about 600 MPa. Comparing the same specimens for TiAlCu_4_ and TiAlCu_5_, they were also increasing flow stress of about 500~600 Mpa with increasing strain rate from 10^−3^ to 4 × 10^3^ s^−1^.

Figure 3a–c present the stress–strain curves of the TiAlCu_x_ specimens tested under dynamic loading conditions. As for the results presented in Figure 2 for the quasistatic loading case, the maximum flow stress increased as the strain rate increased for all of the specimens, while the fracture strain decreased. Furthermore, the specimen with a Cu addition of 4 at.% again possessed the best mechanical properties of the various samples.

### 3.3. Strain Rate Effect

In general, Figure 2 and Figure 3 show that the strain rate has a significant effect on the flow response of the TiAlCu_x_ samples. This effect can be quantified by the strain rate sensitivity factor, β, which is defined as [35]
(1)$$\beta =\partial \sigma /\partial \dot{\varepsilon }=\left(\sigma_{2}-\sigma_{1}\right)/\ln({\dot{\varepsilon }}_{2}/{\dot{\varepsilon }}_{1})$$
where $\sigma_{2}$ and $\sigma_{1}$ are the compressive stresses obtained at average strain rates of $\dot{\varepsilon_{2}}$ and ${\dot{\varepsilon }}_{1}$*,* respectively. Figure 4a–c show the variation of β with the strain at different strain rates for the TiAlCu_x_ samples with Cu additions of 3, 4 and 5 at.%, respectively. It can be seen that for each sample, the strain rate sensitivity increased as the strain and strain rate increased. Furthermore, for a given strain and strain rate, β increased with an increasing Cu content. The effects of the strain rate and Cu content on the strain rate sensitivity reflect the changes in the microstructural evolution of the sample, e.g., the grain size and precipitate formation. The results in Figure 4 indicate that the TiAlCu_x_ sample with 4 at.% Cu addition had the highest strain rate sensitivity of all the samples. The results for the strain rate sensitivity in Figure 4 are thus consistent with those for the stress−strain behaviour in Figure 2 and Figure 3. Overall, the results indicate that, while Cu addition enhances the strength of TiAl intermetallic alloy, the level of Cu addition should be controlled to 4 at.% in order to optimize the microstructure and mechanical properties.

### 3.4. Microstructural Observations and Fracture Analysis

Figure 5a–c present OM images of the matrix microstructures of the as-cast TiAlCu_x_ specimens with Cu additions of x = 3~5 at.%, respectively. It can be seen that the lamellar microstructure (γ + α_2_) became more pronounced with an increasing Cu content. Notably, while a lamellar microstructure enhanced the mechanical strength of TiAl intermetallic alloys, it also increased their brittleness [34]. Figure 6a–f present SEM fractographs of the three TiAlCu_x_ samples under strain rates of 10^−3^ s^−1^ and 4 × 10^3^ s^−1^, respectively. As shown in Figure 6a,b, the fracture surface of the TiAlCu_3_ specimen contained distinct cleavage-like features characteristic of a brittle fracture mode. Moreover, the number of cleavage features increased at a higher strain rate. It means that the fracture strain decreases when specimens suffer high strain rate deformation. A similar tendency was also observed for the sample containing 4 at.% Cu (see Figure 6c,d). However, the fracture surfaces contained fewer cleavage features than those in Figure 6a,b, and hence it is inferred that the sample has higher ductility than the other two Cu content specimens. As shown in Figure 6e,f, the fracture surfaces of the specimen with 5 at.% Cu addition contained a large number of cleavage features. In other words, the TiAlCu_5_ sample had a brittle characteristic, and thus fractured under a lower flow stress and strain than TiAlCu_4_ sample (as shown earlier in Figure 2 and Figure 3).

### 3.5. Corrosion Property Analysis

Figure 7 presents the polarisation curves of the as-cast TiAlCu_x_ specimens. The TiAlCu_x_ specimen with 4 at.% Cu addition had a corrosion current density, I_corr_, of approximately 3.5 × 10^−8^ A/cm^2^. By contrast, the specimen with 3 at.% Cu had an I_corr_ value of 5.4 × 10^−8^ A/cm^2^, respectively. In other words, the anti-corrosion performance of the TiAlCu_x_ sample improved as the Cu addition was increased from 3 to 4 at.%, However, when the Cu content was further increased to 5 at.%, I_corr_ increased to 8.2 × 10^−8^ A/cm^2^. Thus, overall, the results show that the optimal anti-corrosion performance was obtained for the sample with a Cu addition of 4 at.%. It is speculated that the improved anti-corrosion performance is the result of a lower quantity of brittle γ phase and an increased quantity of α_2_ phase [34].

## 4. Conclusions

The results have shown that, for all of the samples, the flow stress and strain rate sensitivity increase at higher strain rates. By contrast, the fracture strain reduces as the strain rate increases. Among all the samples, that with a Cu addition of 4 at.% exhibits the best mechanical response, i.e., a good flow strength and an improved ductility. The microstructural observations have shown that the TiAlCu_x_ specimens have a lamellar microstructure (γ + α_2_). As the Cu content increases from 3 at.% to 4 at.%, the quantity of brittle γ phase decreases, while that of α_2_ phase increases. However, as the Cu content is further increased to 5 at.%, the quantity of γ phase increases once more. The SEM observations showed that the fracture surfaces of all the samples contained distinct cleavage features. The number of cleavage features reduces as the Cu content increases from 3 at.% to 4 at.%, but then increases as the Cu content is increased to 5 at.%. In the electrochemical corrosion tests, the TiAlCu_x_ sample with 4 at.% Cu addition showed the lowest corrosion current density, I_corr_. Thus, overall, the results indicate that the TiAlCu_4_ sample, with a lower amount of γ phase and higher amount of α_2_ phase, has the best mechanical, microstructural and anti-corrosion properties of the considered samples.

## Figures and Tables

**Figure 1 materials-14-05056-f001:**
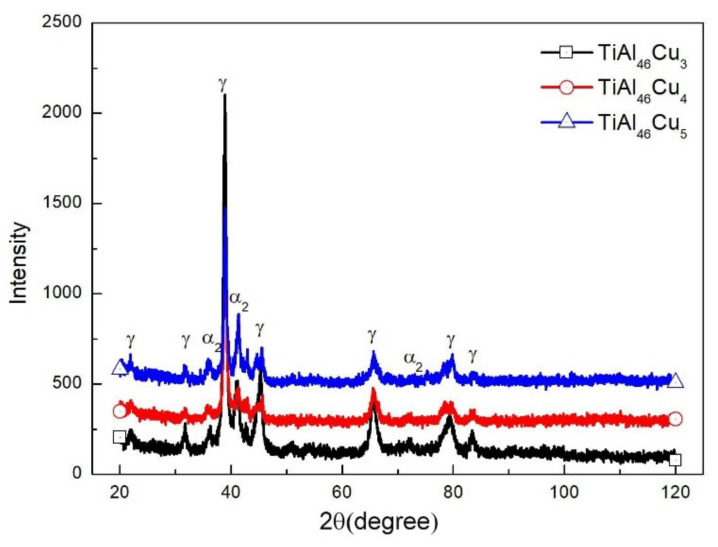
XRD patterns of TiAl_46_Cu_x_ specimens with x = 3, 4 and 5 at.%.

**Figure 2 materials-14-05056-f002:**
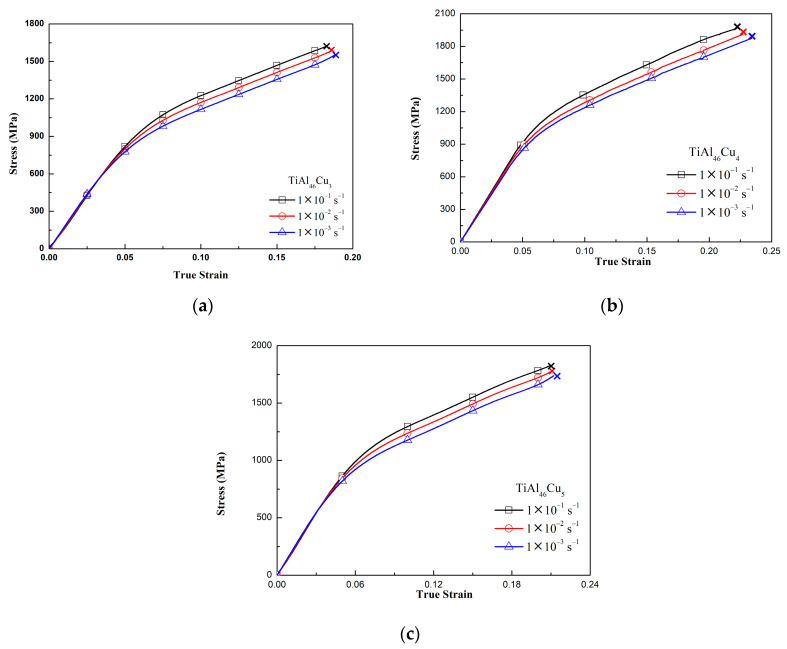
Stress−strain curves of TiAl_46_Cu_x_ specimens under quasistatic strain rates: (**a**) TiAl_46_Cu_3_; (**b**) TiAl_46_Cu_4_; and (**c**) TiAl_46_Cu_5_.

**Figure 3 materials-14-05056-f003:**
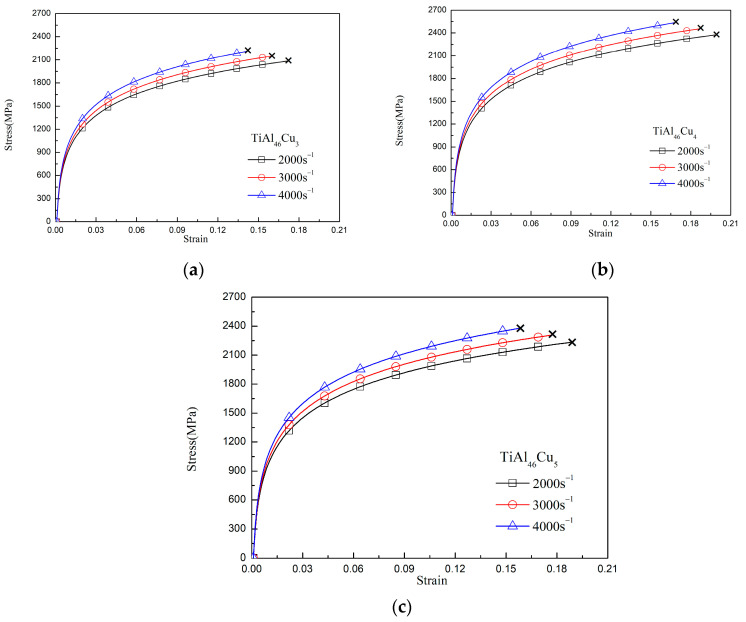
Stress−strain curves of TiAl_46_Cu_x_ specimens under dynamic strain rates: (**a**) TiAl_46_Cu_3_; (**b**) TiAl_46_Cu_4_; and (**c**) TiAl_46_Cu_5_.

**Figure 4 materials-14-05056-f004:**
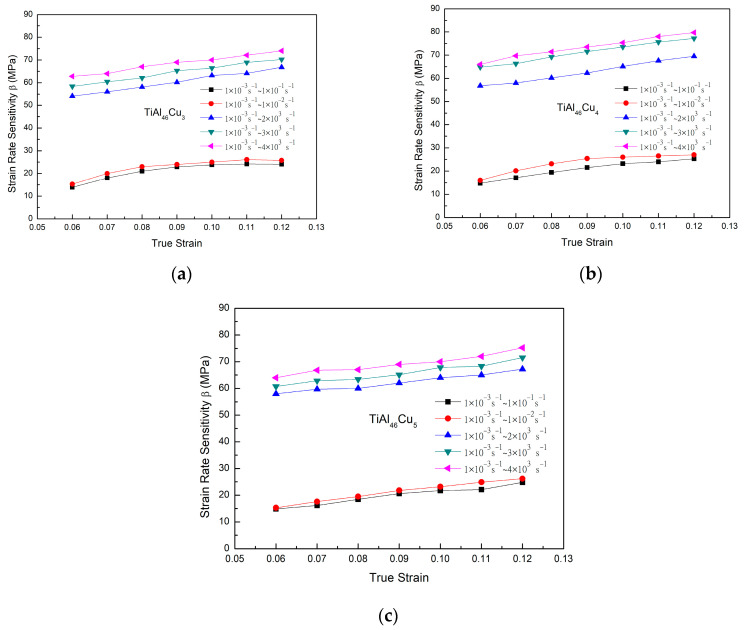
Strain rate sensitivity of TiAl_46_Cu_x_ specimens under dynamic strain rates: (**a**) TiAl_46_Cu_3_; (**b**) TiAl_46_Cu_4_; and (**c**) TiAl_46_Cu_5_.

**Figure 5 materials-14-05056-f005:**
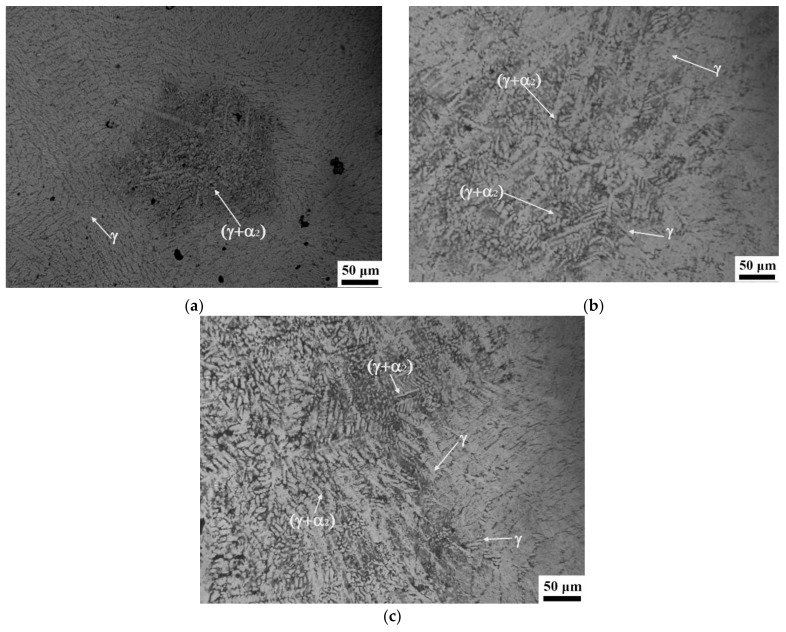
Optical micrographs of as-cast TiAl_46_Cu_x_ specimens: (**a**) TiAl_46_Cu_3_, (**b**) TiAl_46_Cu_4_, and (**c**) TiAl_46_Cu_5_.

**Figure 6 materials-14-05056-f006:**
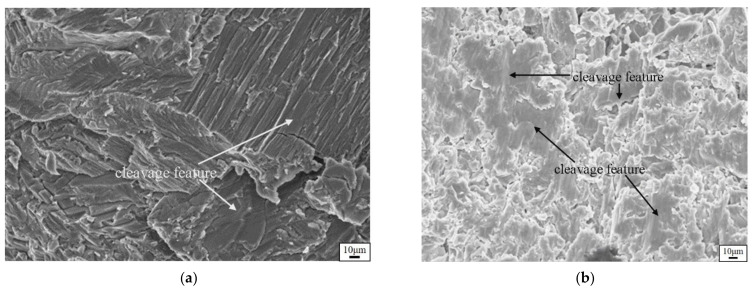
Fracture morphologies of: (**a**) TiAl_46_Cu_3_ specimens deformed at 1 × 10^−3^ s^−1^, (**b**) 4 × 10^−3^ s^−1^; (**c**) TiAl_46_Cu_4_ specimens deformed at 1 × 10^−3^ s^−1^, (**d**) 4 × 10^−3^ s^−1^; (**e**) TiAl_46_Cu_5_ specimens deformed at 1 × 10^−3^ s^−1^, (**f**) 4 × 10^−3^ s^−1^.

**Figure 7 materials-14-05056-f007:**
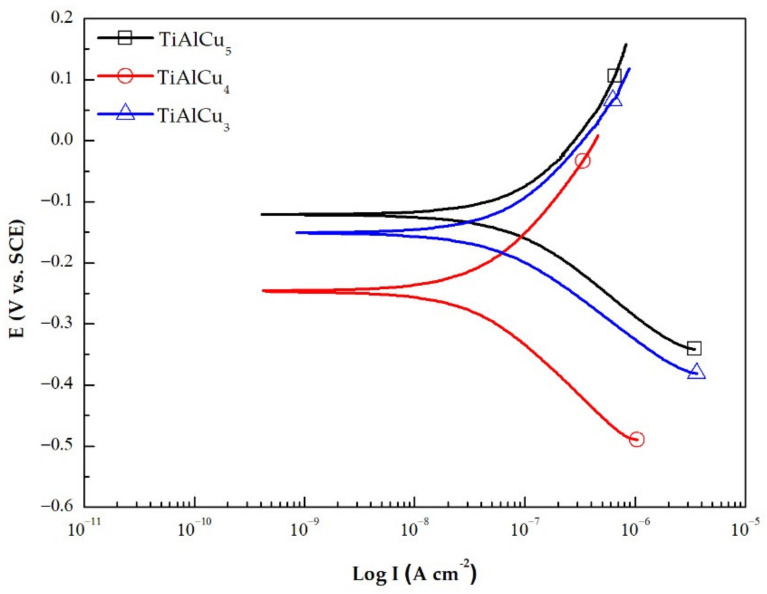
Potentiodynamic polarisation curves of TiAl_46_Cu_x_ specimens with x = 3, 4 and 5 at.%.

**Table 1 materials-14-05056-t001:** The fracture strain for all tested specimens under different strain rates.

	10^−3^	10^−2^	10^−1^	2 × 10^3^	3 × 10^3^	4 × 10^3^
TiAlCu_3_	0.195	0.185	0.18	0.17	0.16	0.14
TiAlCu_4_	0.24	0.23	0.22	0.2	0.19	0.17
TiAlCu_5_	0.22	0.21	0.2	0.19	0.18	0.16

## Data Availability

Not applicable.

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
