# Peer review of "Effects of Cu Addition on Mechanical Behaviour, Microstructural Evolution and Anti-Corrosion Performance of TiAl-Based Intermetallic Alloy under Different Strain Rates"

_materials, 2021, doi:10.3390/ma14175056_

Round 1
Reviewer 1 Report
- In several places, the Greek symbols of phases are lacking: page 1, line 20; page 3, lines 117, 11, 119, and 121.
- Page 1, line 21-23: Among all the specimens, the sample with 4 at.% Cu addition shows the best anti-corrosion performance in Ringer’s solution as a result of the Cu can enhance the anti-corrosion performance fit the TiAl- based intermetallic alloy. This sentence is unclear (as a result of the Cu can enhance…).
- Page 2, line 84: an hour is expressed in the SI system as `h`, not `hr`.
- Page 2, lines 91, 93, 94: 10-3, 10-2 and 10-1 s-1; 3 103 s-1, 4 103 s-1 and 5 103 s-1. The way of writing numbers (exponents) is improper.
- Page 3, line 107-108: (9.0 gL-1 NaCl, 0.43 gL-1 KCl, 0.24 gL-1 CaCl2 and 0.20 gL-1 NaHCO3. It is a lack of subscripts (chemical formula) and superscripts (exponents).
- Page 3, lines 109-110: five polarization curves were obtained -600 mV and 200 mV. The sentence is incorrect (obtained at …).
- Page 3, lines 112-113: the experimental data were collected only once a consistent polarization curve was obtained each time. This sentence is unclear, but the problem is more serious. The expression “consistent” has no meaning; I understand that the authors have considered that the curves were similar, closed to each other. However, if the tests have been repeated, I suggest determining the corrosion potential current density, calculate the means and standard deviations, and give them in the table. If such results do not appear, please anyway calculate the above values and assess the possible physical errors.
- Page 3, lines 123-124: the XRD results suggest that, among the present samples, the TiAl46Cu4 sample has the best mechanical properties. I do not think that such a suggestion is justified and should be deleted.
- Page 4, lines 134-136: the strength of the TiAlCux specimens increase with increasing strain rate and strain. I cannot understand these curves and the reader might have also some difficulties. Please describe in the experimental part the whole procedure in detail. What means `quasi-static` if the samples have been constantly tensed? What strength do you mean if only the first part of the stress-strain curve is shown? Why are the points at the curves, there has been no constant recording for this relationship, but only the measurements of strains at some values of stresses were recorded? And, coming back to the first lines of this comment: I cannot accept such assumption as for me the strain follows the stress what is typical, and where do you see that strength increases with increasing strain? Deformation strengthening? It is no true. And one more thing: without showing us standard deviations in the curves I cannot agree that strength (strengthening) follows the increasing strain rate. Perhaps is it within the limits of an experimental error?
- Page 4, lines 140-143: Figures 3(a)~3(c) present the stress-strain curves of the TiAlCux specimens under 140 dynamic loading conditions. As for the results presented in Fig. 2 for the quasi-static load-141 ing case, the flow stress increases with increasing strain rate for all of the specimens, while 142 the fracture strain decreases. Please show how you determine the flow stress and fracture strain. I suppose that at the end of these curves the samples have broken (but I see no such effects, a fall in the curves) and you mean that you mean that at this point you have maximal flow stress (or rather tensile strength) and fracture strain.
- Page 9, lines 202-206: This study has examined the deformation behaviour and microstructural evolution of TiAlCux intermetallic alloy (x= 3~5 at.%) at strain rates ranging from 1x10—3 ~ 4x103 s-1 and a temperature of 25 C using a material testing system (MTS) and split-Hopkinson pressure bar (SHPB). The electrochemical properties of the various samples have additionally been tested in Ringer’s solution over a voltage range of -600 mV and 200 mV at a rate of 1 mV/s. These two sentences are not conclusions, but a summary of results, and should be deleted.

Author Response
We would very thanks for your efforts. You have made in evaluating this manuscript. Your comments and suggestions are constructive and valuable for us. They will contribute to improving this manuscript. Please find below reply and answers to your questions and comments.
Please see the reply and answers in the attached file.
- In several places, the Greek symbols of phases are lacking: page 1, line 20; page 3, lines 117, 118, 119, and 121.
Reply: Thanks. The Greek symbols of phases are added in page 1 line 21; page 3, line 120 to 125.
- Page 1, line 21-23: Among all the specimens, the sample with 4 at.% Cu addition shows the best anti-corrosion performance in Ringer’s solution as a result of the Cu can enhance the anti-corrosion performance fit the TiAl- based intermetallic alloy. This sentence is unclear (as a result of the Cu can enhance…).
Reply: Thanks. The sentence has modified as “Overall, the results indicate that the favourable properties of the TiAlCu4 sample stem mainly from the low γ phase content of the microstructure and the high α2 phase content.”
- Page 2, line 84: an hour is expressed in the SI system as `h`, not `hr`.
Reply: Thanks. It has modified “hr” to “h”. (in page 2, line 84)
- Page 2, lines 91, 93, 94: 10-3, 10-2 and 10-1 s-1; 3 103 s-1, 4 103 s-1 and 5 103 s-1. The way of writing numbers (exponents) is improper.
Reply: Thanks. All the typing wrongs have modified in the revised version. (in page 2 line 92 and 94)
- Page 3, line 107-108: (9.0 gL-1 NaCl, 0.43 gL-1 KCl, 0.24 gL-1 CaCl2 and 0.20 gL-1 NaHCO3. It is a lack of subscripts (chemical formula) and superscripts (exponents).
Reply: Thanks. The subscript and superscript have modified in page 3, line 109-110.
- Page 3, lines 109-110: five polarization curves were obtained -600 mV and 200 mV. The sentence is incorrect (obtained at …).
Reply: Thanks. The sentence has modified as “using a voltage range of -600 mV to 200 mV and a scan rate of 1 mV/s.” (in page 3 line 113)
- Page 3, lines 112-113: the experimental data were collected only once a consistent polarization curve was obtained each time. This sentence is unclear, but the problem is more serious. The expression “consistent” has no meaning; I understand that the authors have considered that the curves were similar, closed to each other. However, if the tests have been repeated, I suggest determining the corrosion potential current density, calculate the means and standard deviations, and give them in the table. If such results do not appear, please anyway calculate the above values and assess the possible physical errors.
Reply: Thanks. The sentence is confuse to the reviewer and reader. We have modified the sentence as “For each specimen, five polarization curves were obtained under room temperature conditions (28°C ± 1°C) using a voltage range of -600 mV to 200 mV and a scan rate of 1 mV/s. To ensure experimental reliability, data collection was under a similar and standard deviation is within reasonable range (± 10mv), then the polarization curve was obtained, and each alloy was tested five times, with a new sample being used on each occasion.” Furthermore, the corrosion potential current density was calculated by the Tafel extrapolation method and we also did five times experiment for one curve and get the approximately data. The deviation is about ±1´10-9A/cm2.
- Page 3, lines 123-124: the XRD results suggest that, among the present samples, the TiAl46Cu4 sample has the best mechanical properties. I do not think that such a suggestion is justified and should be deleted.
Reply: Thanks. The sentence is deleted in the revised revision.
- Page 4, lines 134-136: the strength of the TiAlCux specimens increase with increasing strain rate and strain. I cannot understand these curves and the reader might have also some difficulties. Please describe in the experimental part the whole procedure in detail. What means `quasi-static` if the samples have been constantly tensed? What strength do you mean if only the first part of the stress-strain curve is shown? Why are the points at the curves, there has been no constant recording for this relationship, but only the measurements of strains at some values of stresses were recorded? And, coming back to the first lines of this comment: I cannot accept such assumption as for me the strain follows the stress what is typical, and where do you see that strength increases with increasing strain? Deformation strengthening? It is no true. And one more thing: without showing us standard deviations in the curves I cannot agree that strength (strengthening) follows the increasing strain rate. Perhaps is it within the limits of an experimental error?
Reply: Thanks for your comment. The quasi-static is meaning that the deformation strain rate is very slow. Comparing our high strain rate testing, the quasi static strain rate of 10-3 s-1 is slower than our high strain rate of 4´103 s-1. So we call that the static strain rate testing is “quasi-static”. It is my mistake and to avoid the confusion, we modified the sentence as “However, the fracture strain reduces as the strain rate increases, which suggests that the TiAlCux specimens undergo a strengthening effect at higher strain rates”. (in line 135-136)
- Page 4, lines 140-143: Figures 3(a)~3(c) present the stress-strain curves of the TiAlCux specimens under dynamic loading conditions. As for the results presented in Fig. 2 for the quasi-static load-ing case, the flow stress increases with increasing strain rate for all of the specimens, while the fracture strain decreases. Please show how you determine the flow stress and fracture strain. I suppose that at the end of these curves the samples have broken (but I see no such effects, a fall in the curves) and you mean that you mean that at this point you have maximal flow stress (or rather tensile strength) and fracture strain.
Reply: Thanks for your comments. The “´” mark in the Figures 2 and 3 mean that the specimens fracture at this point during the deformation. So, we define this point is fracture point and the strain at this point is fracture strain for this testing condition. Furthermore, we mean that the maximum flow stress is at this point. During our testing, we can observe that the maximum flow stress is increasing with increasing strain rate.
- Page 9, lines 202-206: This study has examined the deformation behaviour and microstructural evolution of TiAlCux intermetallic alloy (x= 3~5 at.%) at strain rates ranging from 1x10—3 ~ 4x103 s-1 and a temperature of 25 C using a material testing system (MTS) and split-Hopkinson pressure bar (SHPB). The electrochemical properties of the various samples have additionally been tested in Ringer’s solution over a voltage range of -600 mV and 200 mV at a rate of 1 mV/s. These two sentences are not conclusions, but a summary of results, and should be deleted.
Reply: Thanks. The sentence has deleted

Reviewer 2 Report
Dear Authors,
The subject of the article is up-to-date and fits in with the industry development trends. Introduction is spelled correctly. An analysis of the current issue was carried out and conclusions were drawn on the basis of which the authors decided to strengthen the intermetal Alti with the addition of Cu in the range of 3-5%.The process of preparing the material for research was carried out correctly. Additionally, XRD material was analyzed. The methodology of examining the mechanical properties of quasi-static and dynamic tests is correct.
The microstructure observations are made correctly but can be improved:
- Please provide the scale in the photos with number 5 and the marking of the microstructure description in the photos.
The fracture analysis of microfractive fractures is presented briefly.
- I am asking for a broader analysis of photos 6 a-f. Please, mark in the photos the specific characteristic cracking mechanisms
Corrosion analysis results are correct.
The conclusions resulting from the research are correct.
The article is at a high level of merit, but the indicated comments will significantly improve its quality.
Best Regards
Author Response
We would very thanks for your efforts. You have made in evaluating this manuscript. Your comments and suggestions are constructive and valuable for us. They will contribute to improving this manuscript. Please find below reply and answers to your questions and comments.
Please see the reply and answers in the attached file.
The microstructure observations are made correctly but can be improved:
- Please provide the scale in the photos with number 5 and the marking of the microstructure description in the photos.
The fracture analysis of microfractive fractures is presented briefly.
Reply: Thanks. We have added the scale and the marking of the microstructure.
- I am asking for a broader analysis of photos 6 a-f. Please, mark in the photos the specific characteristic cracking mechanisms
Reply: Thanks. We have mark the specific characteristic crack structure in the Figs. 6a-6f in the revised version. And we modified the description of fracture analysis for the photos 6 a-f in the revised version.
Corrosion analysis results are correct.
Reply: Thanks your comments.
The conclusions resulting from the research are correct.
The article is at a high level of merit, but the indicated comments will significantly improve its quality.
Reply: Thanks your comments.

Reviewer 3 Report
In this paper, the authors studied the mechanical behaviour, structure and corrosion behaviour of TiAl intermetallic with various Cu addition. Despite of lots of content presented here, I do not see any novelty in this work. Furthermore, too many typos and errors can be found in the paper, which means the author do not carefully examine their results and proof-reading. Therefore, this work is far away from the criteria of Materials. And I think this work can not be published in Materials. My comments are as follows:
- The authors should proof read the paper very carefully, there are lots of typos and errors in the paper. For instance, line 20, line 91, line 93, line 117-120 etc. The x-axis in Figure 1 is also completely wrong. From these errors, I don't believe the authors take this work seriously.
- From line 131-133, the authors claim the strain rate has a greater effect on flow stress than the strain. However, the authors do not provide the quantitative analysis regarding strain state and strain. In addition, the authors believe the fracture strain decreases with increasing strain rate from Figure2. However, we can definitely see there is very tiny difference among fracture strain (almost the same), which means that the strain rate might play negligible effect on fracture strain.
- In line 160-162, the authors claim the strain rate sensitivity can be attributed to microstructural evolution. But here the authors do no provide high quantity characterization of microstructure such as TEM and HRTEM. No strong evidence can be discovered to support this point.
Author Response
We would very thanks for your efforts. You have made in evaluating this manuscript. Your comments and suggestions are constructive and valuable for us. They will contribute to improving this manuscript. Please find below reply and answers to your questions and comments.
Please see the reply and answers in the attached file. Thanks.
- The authors should proof read the paper very carefully, there are lots of typos and errors in the paper. For instance, line 20, line 91, line 93, line 117-120 etc. The x-axis in Figure 1 is also completely wrong. From these errors, I don't believe the authors take this work seriously.
Reply: Thanks for your comments. This is my negligence, the typos and x-axis of Figure 1 have modified in the revised revision. (in lines 21, 91-94, 120-124 and Figure 1)
- From line 131-133, the authors claim the strain rate has a greater effect on flow stress than the strain. However, the authors do not provide the quantitative analysis regarding strain state and strain. In addition, the authors believe the fracture strain decreases with increasing strain rate from Figure2. However, we can definitely see there is very tiny difference among fracture strain (almost the same), which means that the strain rate might play negligible effect on fracture strain.
Reply: Thanks for your comments. The effect of strain rate on the maximum flow stress is more than fracture strain, we can see the Figure 2a and Figure 3a, the maximum flow stress of TiAlCu3 is increasing from 1500MPa at strain rate of 10-3 s-1 to 2100 MPa at strain rate of 4´103 s-1. The flow stress is increasing about 600MPa. Comparing the same specimens for TiAlCu4 and TiAlCu5, they are also increasing flow stress of about 500~600MPa with increasing strain rate from 10-3 to 4´103 s-1. So the strain rate has greater effect on the flow stress. However, the strain rate also has effect on fracture strain (see Table 1). As the reviewer mentioned, the strain rate just a tiny effect on the fracture strain and we have modified the description in the revised version.
- In line 160-162, the authors claim the strain rate sensitivity can be attributed to microstructural evolution. But here the authors do no provide high quantity characterization of microstructure such as TEM and HRTEM. No strong evidence can be discovered to support this point.
Reply: Thanks. The microstructure analysis of TEM could help us to realize the microstructural evolution on the effect of strain rate. However, we also can investigate the fracture feature by SEM and the microstructure by OM to analyze the strain rate effect. Due to the phase observation by OM, it can find generation of specific phase under different strain rate. Furthermore, the fracture feature was observed by SEM, it also can investigate the strain rate effect. Therefore, this study utilizes the two microstructural observation technologies

Reviewer 4 Report
This manuscript addresses the effects of Cu addition on the mechanical behavior, microstructural evolution, and anti-corrosion performance of TiAl-based intermetallic alloys. The experimental results suggested that the sample with 4 atomic percent Cu addition exhibited the best anti-corrosion performance.
1. There were a few typographical errors, such as the numbers not being in subscripts on lines 91 and 93.
2. More detail is needed in the experimental section in order to help the reader possibly reproduce the experiments. Such detail is lacking in the description of the X-ray diffraction experiments. Please list parameters such as the X-ray and scattering angles.
3. In the X-ray diffraction results (Section 3.1), some detail in missing in the discussion such as what type of phase is present as there are blank spaces where I think the phase should be. Please address this issue. Furthermore, was quantitative phase analysis performed? I am not sure how the authors can come to their conclusion based on there being no quantitative results.
4. There are also some issues with the figures. For example, where are the scale bars in Figure 5? The authors should probably label the lamellar microstructure in the figure as well.
5. Also, the scale bars are very hard to read in Figure 6. It is recommended that the authors draw there own scale bars on the figures to improve the visibility.
6. Where did the authors find Equation (1)? Please provide a reference.
7. In Section 3.2, please quantify the fracture strain for all the samples.
8. Why was there a large increase in the strain rate sensitivity when the strain rate increased from 1 x 10^-3 - 1 x 10^-2 s^-1 to 1 x 10^-3 - 2 x 10^-3 s^-1?
9. Please spell out abbreviations, such as at.%, in the abstract.
Based on these comments, I must recommend major revisions.
Author Response
We would very thanks for your efforts. You have made in evaluating this manuscript. Your comments and suggestions are constructive and valuable for us. They will contribute to improving this manuscript. Please find below reply and answers to your questions and comments.
Please see the reply and answers in the attached file. Thanks.

Round 2
Reviewer 3 Report
In second round, the authors took my suggestions to carefully check the paper and correct the typos and errors. The authors appropriately answered my questions. I think it is suitable for publication now.
Reviewer 4 Report
The revisions look adequate, although it looked confusing as some things that were added were also crossed off as if they had been deleted.